# Cell-Free Supernatant from *Lactobacillus* and *Streptococcus* Strains Modulate Mucus Production via Nf-κB/CREB Pathway in Diesel Particle Matter-Stimulated NCI-H292 Airway Epithelial Cells

**DOI:** 10.3390/molecules28010061

**Published:** 2022-12-21

**Authors:** Ji Yeon Lee, Chang-Ho Kang

**Affiliations:** MEDIOGEN, Co., Ltd., Biovalley 1-ro, Jecheon-si 27159, Republic of Korea

**Keywords:** airway inflammation, cAMP response element-binding protein, *Lactobacillus*, *Streptococcus*

## Abstract

Airway epithelial cells are a major site of airway inflammation and may play an important role in the pathogenesis of chronic obstructive pulmonary disease (COPD). Diesel particulate matter (DPM) is associated with mucus hypersecretion and airway inflammation and has been reported to overexpress airway mucin in the NCI-H292 airway epithelial cells. Therefore, regulation of mucin hypersecretion is essential for developing novel anti-inflammatory agents. This study aimed to investigate the effects of cell-free supernatant (CFS) from *Lactobacillus* and *Streptococcus* on nitro oxide (NO) production in RAW264.7 and proteins associated with mucus production in NCI-H292 cells. We observed that NO production was reduced by CFS from *Lactobacillus* and *Streptococcus* in RAW 264.7, and *MUC4*, *MUC5AC*, and *MUC5B* gene expression was increased by phosphorylation of nuclear factor kappa B (NF-κB) p65 and cAMP response element-binding protein (CREB) in DPM-stimulated NCI-H292 cells. However, CFS from *L. paracasei* MG4272, MG4577, *L. gasseri* MG4247, and *S. thermophilus* MG5140 inhibited mRNA expression related to mucus production by downregulating the CREB/NfκB signaling pathway. These results suggest *that* CFS from *L. paracasei* MG4272, MG4577, *L. gasseri* MG4247, and *S. thermophilus* MG5140 can contribute as a strategic candidate to the prevention of airway inflammatory diseases caused by DPM.

## 1. Introduction

Airway inflammation is a major cause of chronic obstructive pulmonary disease (COPD), a non-communicable disease that causes pulmonary and extra-pulmonary symptoms [1]. Inhalation toxicity pollutants cause chronic airway inflammation by activating epithelial cells and macrophages in the lungs, and systemic inflammation occurs when these inflammatory mediators are exposed to the circulatory system [2,3]. Airway inflammation increases the production of a mixture of sputum by excess mucin and other glycoproteins [4]. Mucin production is regulated by nine membrane-tethered genes and seven gel-forming genes. Among them, *MUC5AC* and *MUC5B*, which are gel-forming genes, account for 90% of mucins [5]. Activation of the mucin transcription factor mainly involves the expression of the nuclear factor kappa B (NfκB) p65/cAMP response element-binding protein (CREB) pathway via phosphorylation of extracellular signal-regulated kinase (ERK) 1/2 and p38 mitogen-activated protein kinases (p38) [6,7,8]. COPD is one of the five leading causes of death worldwide, with an increase in the prevalence of 15.6% between 2007 and 2017 [9]. According to recent reports, the mortality rate has been increasing due to respiratory diseases, such as COPD caused by DPM, in recent years [10]. There are various drug therapies to treat airway inflammation such as COPD, but the early mortality rate is high because it imposes a significant economic burden [11]. Thus, there is a need for research on alternative therapies that have been molecularly proven to function in airway inflammation.

Probiotics, including the bacterial genera *Lactobacillus*, *Bifidobacterium*, *Streptococcus,* and *Enterococcus*, are living microorganisms that have beneficial effects on human health and are generally considered safe [12]. Probiotics are known to have beneficial effects on human health by forming a healthy microbiome and improving intestinal barrier function in the human intestine [13]. Recently, probiotics had various beneficial effects, such as preventing respiratory viruses, including COVID-19, anti-allergy, improvement of metabolic disease, and anti-inflammation on the skin [14,15,16]. Among the probiotic species, *Lactocaseibacillus rhamnosus*, *Bifidobacterium breve*, and *Escherichia coli* Nissle 1917 have been reported to prevent inflammation in a COPD mouse model but have not been studied in other species [17,18]. In addition, a probiotic suppression mechanism for inflammation and mucin generation in airway epithelial cells has not yet been established. In vitro evaluation has advantages such as more economical, direct access and investigation to cellular components, easier and quicker to perform, and reduction in the number of animals used in research [19]. Moreover, as a cell model for the evaluation of COPD, commonly used NCI-H292 airway epithelial cells from human lungs, which are similar to the response of airway epithelium to cigarette smoke in vivo [19]. To induce airway inflammation among various agents, including cigarette smoke extract, fly ash, lipopolysaccharides, and carbon black, diesel exhaust particles (DPM) have been reported to induce mucin and inflammation in NCI-H292 airway epithelial cells [20]. Based on this considerable evidence, it was recently reported that the airway inflammation of saponin was confirmed using an in vitro model in which DPM was treated with NCI-H292 [21].

In this study, we confirmed that inhibition of cell-free supernatant (CFS) from *Lactobacillus* and *Streptococcus* strains on inflammation via NO production in LPS-induced RAW264.7 cells. Moreover, the effect of CFS from Lactobacillus and Streptococcus strains on the expression of mucin-related genes and proteins in NCI-H292 cells treated with DPM was investigated.

## 2. Results

### 2.1. Inhibition of NO Production by CFS from Lactobacillus and Streptococcus Strains in LPS-induced RAW 264.7 Cells

The cytotoxicity of 2.5 and 5% CFS in RAW 264.7 was established by MTT assay (Figure 1A). The results showed that 5% CFS of *Lactobacillus* and *Streptococcus* strains showed no cytotoxicity (≥100%) in RAW 264.7 cells. Moreover, LDH release of 5% CFS on LPS-induced RAW 264.7 cells were measured, shown in Figure 1B. All CFS decreased LDH release compared to LPS-induced control. Therefore, NO production was evaluated at 5% CFS, a concentration without cytotoxicity. As shown in Figure 1C, the CFS of all probiotics remarkably inhibited NO production compared to the LPS-induced controls.

### 2.2. CFS from Lactobacillus and Streptococcus Strains Suppress Cytotoxicity in DPM-induced NCI-H292 Cells

First, cell morphology was observed to set the treatment time of DPM on NCI-H292 cells (Figure 2A). As a result, it was confirmed that DPM completely penetrated the NCI-H292cells after 24 h of treatment, so subsequent experiments were conducted at 24 h treatment. Based on the results of RAW 264.7 cells, the cytotoxicity of 5 and 10% CFS in NCI-H292 cells was assessed by MTT assay. In Figure 2B,C, 5% CFS of *Lactobacillus* and *Streptococcus* strains showed no cytotoxicity (≥86.74%) in the presence or absence of DPM for 24 h. Moreover, LDH release was measured at 5% CFS. As a result, the 5% CFS of all probiotics significantly inhibited LDH release compared to the DPM-induced controls.

### 2.3. Inhibition of mRNA Expression Involved in Mucus Production by CFS from Lactobacillus and Streptococcus Strains in DPM-induced NCI-H292 Cells

The effect of CFS from *Lactobacillus* and *Streptococcus* strains on mRNA expression related to mucus production was confirmed in DPM-induced NCI-H292 cells. It was found that treatment with DPM upregulated all mRNA expressions, whereas treatment with *L. paracasei* MG4272, MG4577, *L. gasseri* MG4247, and *S. thermophilus* MG5140 significantly suppressed *MUC4* (0.11-, 0.22-, 0.11-, and 0.21-fold), *MUC5AC* (0.07-, 0.14-, 0.13-, and 0.15-fold), and *MUC5B* (0.10-, 0.17-, 0.09-, and 0.02-fold) mRNA expression in NCI-H292 cells (Figure 3).

### 2.4. CFS Lactobacillus and Streptococcus Strains Modulate CREB Signaling Pathway in DPM-induced NCI-H292 Cells

To determine the time-dependent effect of each protein, NCI-H292 cells were incubated with DPM. Maximum induction of p-ERK1/2, p-p38 MAPK, and p-CREB protein expression by DPM was observed at 30 min (Figure 4A). The CFS of *Lactobacillus* and *Streptococcus* strains was pretreated for 1 h and treated with DPM for 30 min to confirm the expression of each protein in NCI-H292 cells. It was demonstrated that *L. paracasei* MG4272, MG4577, *L. gasseri* MG4247, and *S. thermophilus* MG5140 significantly decreased the expression of p-p38 (0.75, 0.69, 0.48, and 0.42-fold), p-ERK1/2 (0.39, 0.68, 0.44, and 0.58-fold), and p-CREB (0.69, 0.68, 0.78, and 0.61-fold; Figure 4B).

### 2.5. Effect on Phosphorylation of Nf-κB by CFS from Lactobacillus and Streptococcus Strains in DPM-induced NCI-H292 cells

Protein expression of NF-κB phosphorylation in NCI-H292 cells treated with DPM at various time points was measured using western blotting (Figure 5A). Before 8 h of treatment with DPM, there was no change in NF-κB phosphorylation in NCI-H292 cells; however, at 18 and 24 h, there was a gradual increase. As shown in Figure 5B, treatment with CFS *Lactobacillus* and *Streptococcus* strains reduced the phosphorylation of NF-κB, except for MG4604, in DPM-induced NCI-H292 cells. *L. paracasei* MG4272, MG4577, *L. gasseri* MG4247, and *S. thermophilus* MG5140 markedly inhibited the protein expression of p-NF-κB (0.54-, 0.57-, 0.32-, and 0.49-fold, respectively) in DPM NCI-H292 cells.

## 3. Discussion

COPD is an airway inflammatory disease caused by genetic predisposition, smoking, environmental factors such as DPM and cigarette smoke, and chronic inflammation. Immune responses play an important role in the progression of COPD [3]. In COPD, although probiotics are unlikely to act directly in the lungs via the circulatory system, systemically, metabolites from probiotics modulate inflammatory factors to activate immune cells and macrophages, thereby indirectly protecting against airway inflammation [13]. Metabolites, short-chain fatty acids (SCFA), and microbe-associated molecular patterns (MAMPs) derived from the probiotics suppress the loss of immune homeostasis and inflammatory response caused by excessive DPM exposure by promoting tight-junction proteins to repair the intestinal epithelial barrier function or prevents COPD by directly influencing pulmonary immune homeostasis [22]. Recently, it was reported that metabolites of probiotics carried through the circulation to the lungs might help with COVID-19 by inhibiting viral replication or enhancing the immune response [23]. This inhibition of inflammation has been demonstrated in animal studies that induced COPD with DPM and cigarette smoke; *Bifidobacterium breve* and *Lactobacillus rhamnosus* reduced gene expression related to mucin production and inflammatory factors, NF-κB, and cytokines [18,24]. *Lactobacillus casei* HY2782 and *Bifidobacterium lactis* HY8002 decreased oxidative stress and pro-inflammatory cytokines in the lung in vivo [25]. In addition, since DPM has been reported to cause intestinal imbalance, probiotics may contribute to the prevention of airway inflammation by altering the intestinal microflora [26]. In this study, we attempted to prove the efficacy of CFS from *Lactobacillus* and *Streptococcus* strains against inflammatory factors in macrophages, mucin production, and airway inflammation in NCI-H292 airway epithelial cells induced by DPM. Based on these results, it was intended to be presented as a candidate probiotic with COPD prevention efficacy.

As mentioned above, systemic inflammation can affect airway inflammation in the respiratory system [1]. LPS-induced RAW264.7, a macrophage-like cell line, is an in vitro model that can confirm systemic inflammation [27]. In our results, NO production in LPS-induced RAW 264.7 was decreased by 5% CFS from *Lactobacillus* and *Streptococcus*. NO plays a vital role in the immune response to inflammatory activity and is beneficial for the host’s defense against pathogens and parasites [28]. Additionally, *Lactobacillus* and *Streptococcus* reduced pro-inflammatory cytokines, interleukin 6 (*IL-6*), and tumor necrosis factor (*TNF*)*-α* in our previous study and Appendix A [29]. TNF and IL6 are recognized as major factors in the systemic inflammatory immune response, and overexpression of these factors leads to respiratory diseases, including COPD and COVID-19 [30,31]. These results reveal that CFS of *Lactobacillus* and *Streptococcus* can indirectly reduce airway inflammation by regulating various systemic inflammatory factors in macrophages.

In inflammatory airway diseases, such as COPD, cough and sputum symptoms due to excessive mucus secretion appear [32]. In 11 mucus samples mainly expressed in the lungs, *MUC5AC* and *MUC5B* mRNA are upregulated by inflammatory cytokines or DPM as major factors that produce airway mucus [33]. In addition, it has been reported that the expression of *MUC4* mRNA can affect airway inflammation in human airway epithelial cells [5]. Our results showed that *L. paracasei* MG4272, MG4577, *L. gasseri* MG4247, and *S. thermophilus* MG5140 inhibited the mRNA expression of *MUC4*, *MUC5AC*, and *MUC5B*, which are mucin-producing genes, in NCI-H292 cells treated with DPM. Phosphorylation of CREB by p-ERK1/2 and/or p-p38 MAPK is a major intracellular mechanism for the expression *of MUC4*, *MUC5AC*, and *MUC5B* genes [8,34]. It has been reported that DPM-induced mucin production is mainly mediated by ERK and p38 MAPK [8]. In our results, the phosphorylation of ERK1/2 and p38 was significantly suppressed by *L. paracasei* MG4272, MG4577, *L. gasseri* MG4247, and *S. thermophilus* MG5140 CFS in DPM-induced NCI-H292 cells. In addition, DPM elevates NF-κB signaling pathways, which is an inflammatory/immune response mediator in human airway epithelial cells, and overexpression of NF-κB phosphorylation activated by p-p38 results in mucin overproduction [6,35]. The CFS from *L. paracasei* MG4272, MG4577, *L. gasseri* MG4247, and *S. thermophilus* MG5140 markedly reduced NF-κB phosphorylation in DPM-induced NCI-H292 cells. *L. paracasei* MG4272, MG4577, *L. gasseri* MG4247, and *S. thermophilus* MG5140, which were proven to inhibit mucus and anti-inflammatory effects in this study, can be used as safe probiotics because their hemolysis, cytotoxicity, and adhesion to intestinal epithelial cells have been confirmed [29]. In our previous reports, *L. paracasei* MG4272, MG4577, and *L. gasseri* MG4247 attenuate allergic inflammatory response in mast cells by modulating signal transducer and activator of transcription 6 (STAT6) phosphorylation [29]. Thus, these strains may effectively modulate allergic reactions and airway inflammation related to the respiratory tract.

In summary, CFS from *L. paracasei* MG4272, MG4577, *L. gasseri* MG4247, and *S. thermophilus* MG5140 downregulated mucin production by indirectly reducing pro-inflammatory factors in macrophages and directly inhibiting airway inflammation by modulating the CREB/Nf-κB signaling pathway, which is involved in the expression of *MUC4*, *MUC5AC,* and *MUC5B* in NCI-H292 airway epithelial cells treated with DPM. It seems that *L. paracasei* MG4272, MG4577, *L. gasseri* MG4247, and *S. thermophilus* MG5140 are potential functional foods that can prevent COPD-related airway inflammation; however, this must be verified through additional animal and clinical studies.

## 4. Materials and Methods

### 4.1. Chemicals and Reagents

De Man, Rogosa, and Sharpe (MRS) broth were obtained from BD Biosciences (Franklin Lakes, NJ, USA). For cell culture, RPMI1640, Dulbecco’s modified Eagle’s medium (DMEM), fetal bovine serum (FBS), 100 U/mL penicillin, and 100 U/mL streptomycin (P/S) were purchased from Gibco (Gaithersburg, MD, USA). Diesel particulate matter (DPM) was purchased from the National Institute of Standards and Technology (NIST, Gaithersburg, MD, USA). To investigate the mRNA expression, NucleoZol (MACHEREY-NAGEL, Gutenberg, Hoerdt Cedex, France), Maxime™ RT PreMix (iNtRON, Seongnam-si, Gyeonggi-do, Republic of Korea), and AmfiSure qGreen Q-PCR Master Mix (Gendepot, Katy, TX, USA) were used. Radioimmunoprecipitation assay (RIPA) cell lysis buffer, Xpert Duo inhibitor cocktail solution, anti-mouse and rabbit horseradish peroxidase (HRP)-conjugated secondary antibodies, Bradford (Coomassie) reagent, and West-Q Femto Clean enhanced chemiluminescence (ECL) solution were obtained from Gendepot (Katy, TX, USA). MOPS running buffer and 8 and 10% Bis-Tris gels were obtained from GeneSTAR (Shanghai, China). Extracellular signal-regulated kinases (ERK), phosphorylated (p)-ERK, p38, p-p38, p-nuclear factor kappa B (NfκB) p65 (Cell Signaling Technology, Danvers, MA, USA), NF-κB p65, cAMP response element-binding protein (CREB), p-CREB, and β-actin (Santa Cruz, CA, USA) were used as primary antibodies. Smart-Block™ 5 min-Fast Blocking buffer was obtained from Biomax (Seoul, Republic of Korea). All the reagents not listed above were purchased from Sigma-Aldrich (St. Louis, MO, USA).

### 4.2. Preparation of Cell-Free Supernatants (CFS) from Lactobacillus and Streptococcus Strains

*L. plantarum* MG4064, MG4555, MG5087, *L. paracasei* MG5015, MG4272, MG4577, *L. gasseri* MG4247, and *S. thermophilus* MG5140 provided by MEDIOGEN (Jecheon, Republic of Korea) were cultured in MRS broth in an anaerobic chamber at 37 °C. After 18 h, the mixture was centrifuged (4000× *g*) for 15 min at 4 °C. The supernatant was collected by filtration using a 0.2 μm polytetrafluoroethylene (PTFE) membrane (ADVANTEC, Tokyo, Japan) and used as the CFS. The isolates of each strain were of human origin (MG4064, MG4555, MG4272, MG4577, and MG4247) and fermented foods (MG5087, MG5015, and MG5140). All probiotic strains were identified using 16S rRNA gene sequencing (SolGent Co., Ltd., Daejeon, Republic of Korea), and the DNA sequences were registered in the National Center for Biotechnology Information (NCBI) database using the Basic Local Alignment Search Tool (MG4064, OP102458.1; MG4555, MN400218.1; MG5087, OP102515.1; MG5015, OP102473.1; MG4272, MW947164.1; MG4577, MN833017.1; MG4247, MN069036.1; MG5140, MN055727.1).

### 4.3. Cell Culture

The RAW264.7 cell (American Type Culture Collection, Manassas, VA, USA) were cultured in DMEM containing 10% heat-inactivated FBS and 1% P/S. NCI-H292 airway epithelial cells (Korean Cell Line Bank, Seoul, Republic of Korea) were cultured in RPMI1640 medium with 10% heat-inactivated FBS and 1% P/S. The cells were maintained at 37 °C in a 5% CO2 incubator and passaged after cells reached 70–80% confluency.

### 4.4. Cell Viability

A 3-(4,5-Dimethyl-2-thiazolyl)-2,5-diphenyl-2H-tetrazolium bromide (MTT) assay was used to determine cell viability [36]. The RAW 264.7 and NCI-H292 cells were aliquoted at a density of 2 × 10^4^ cells/well in 96-well plates. After incubation for 24 h, the CFS from *Lactobacillus* and *Streptococcus* strains and MRS (as control) was pretreated for 1 h and then co-treated with LPS (0.5 μg/mL) or DPM (20 μg/mL) for 24 h. After the cells were incubated with the MTT solution (0.1 mg/mL) for 2 h, the MTT solution was removed, and the formazan crystals were lysed using DMSO (150 µL). Absorbance (550 nm) was recorded using a microplate reader (Biotek, Winooski, VT, USA).

### 4.5. Assessment of NO Production

As previously described, NO production was measured using Griess reagent [37]. The RAW264.7 cells were seeded in 24-well plates (3 × 10^5^ cells/mL). After treatment with CFS from *Lactobacillus* and *Streptococcus* strains and MRS (as control) for 1 h and treatment with LPS (0.5 μg/mL) for 24 h, the cell culture supernatant (50 μL) was mixed with Griess reagent (50 μL), followed by measurement at 550 nm with a microplate reader (BioTek).

### 4.6. Preparation of mRNA and Real-Time Polymerase Chain Reaction (qRT-PCR)

RAW 264.7 (7 × 10^5^ cells/well) and NCI-H292 cells (4 × 10^5^ cells/well) were seeded in 6-well plates. Cells were pretreated with CFS from *Lactobacillus* and *Streptococcus* strains and MRS (as control) for 1 h and then treated with LPS (0.5 μg/mL) in RAW 264.7 and/or DPM (20 μg/mL) in NCI-H292 cells. The purity of mRNA was measured using μDrop plates (Thermo Scientific, Waltham, MA, USA) and a microplate reader (BioTek). mRNA was isolated using NucleoZol, following the manufacturer’s protocols. cDNA was prepared using the isolated mRNA and Maxime RT PreMix. qRT-PCR was performed using the CFX96™ System (Bio-Rad, Hercules, CA, USA) with AmfiSure qGreen Q-PCR Master Mix. The primer sequences used are listed in Table 1 and Appendix A. Relative quantitative expression was analyzed using the 2^−ΔΔCT^ method and normalized using the housekeeping gene glyceraldehyde-3-phosphate dehydrogenase (*GAPDH*) [38]. The relative mRNA expression was determined as the fold-change of LPS- or DPM-treated control.

### 4.7. Protein Extraction

For total protein extracts, cells were pretreated with CFS from *Lactobacillus* and *Streptococcus* strains and MRS (as control) for 1 h and then treated with LPS (0.5 μg/mL) in RAW 264.7 and/or DPM (20 μg/mL) in NCI-H292 cells. Protein lysates were obtained by RIPA lysis buffer containing phosphatase and protease inhibitors. Lysates were centrifuged at 13,000 rpm for 15 min at 4 °C, and the supernatants containing the extracted proteins were collected and stored at −80 °C. The extracted proteins were quantified at 1 μg/μL using Bradford reagent (Coomassie). The protein sample was prepared by diluting in 4X LDS sample buffer and heating at 70 °C for 10 min.

### 4.8. Western Blotting

Western blotting was performed as previously reported [39]. Briefly, protein samples were loaded onto 8 and 10% Tris–Bis gels and electrophoresed in MOPS buffer. Proteins were transferred to polyvinylidene difluoride (PVDF) membranes (Millipore, Middlesex County, MA, USA) and washed with TBS-Tween buffer (TBST). After blocking with Smart-Block™ 5 min Fast Blocking Buffer, the membranes were incubated with primary antibodies (1:1000) overnight at 4 °C. After washing three times with TBS-Tween buffer; the membranes were incubated with HRP-conjugated secondary antibodies (1:5000) for 1 h. The membrane was developed using LuminoGraph III Lite (ATTO, Tokyo, Japan) with West-Q Femto Clean ECL solution, and densitograph analysis was performed using CS Analyzer 4(ATTO).

### 4.9. Statistical Analysis

Results are expressed as mean ± standard error of the mean (SEM). One-way analysis of variance (ANOVA) and Tukey’s multiple comparison test (GraphPad Software, Inc., San Diego, CA, USA) was considered statistically significant at *p* < 0.05.

## 5. Conclusions

In the present study, CFS from *L. paracasei* MG4272, MG4577, *L. gasseri* MG4247, and *S. thermophiles* MG5140 was effective in modulating inflammation in RAW264.7 and NCI-H292 airway epithelial cells. We demonstrated that *L. paracasei* MG4272, MG4577, *L. gasseri* MG4247, and *S. thermophilus* MG5140 decreased mRNA expression related to mucus production by attenuating the CREB/NF-κB signaling pathway in DPM-induced airway epithelial cells (Figure 6). In conclusion, *L. paracasei* MG4272, MG4577, *L. gasseri* MG4247, and *S. thermophiles* MG5140 may be used as safe dietary supplements to improve airway inflammation, which needs to be proven in further animal studies.

## Figures and Tables

**Figure 1 molecules-28-00061-f001:**
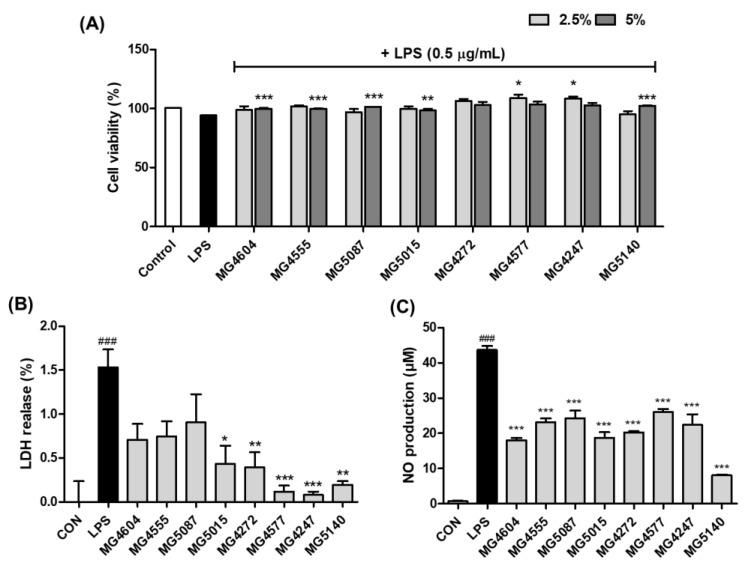
Effect of CFS from *Lactobacillus* and *Streptococcus* strains on cell viability with LPS (**A**), lactate dehydrogenase (LDH) release (**B**), and production of nitro oxide (NO) with LPS (**C**) in RAW 264.7 cells. The RAW 264.7 cells were pretreated with 2.5 and/or 5% of CFS for 1 h and then incubated with or without only LPS (0.5 µg/mL) for 24 h. The results indicate the mean ± SEM of three separate experiments. ### *p* < 0.001 compared with control and * *p* < 0.05, ** *p* < 0.01, and *** *p* < 0.001 compared with LPS alone.

**Figure 2 molecules-28-00061-f002:**
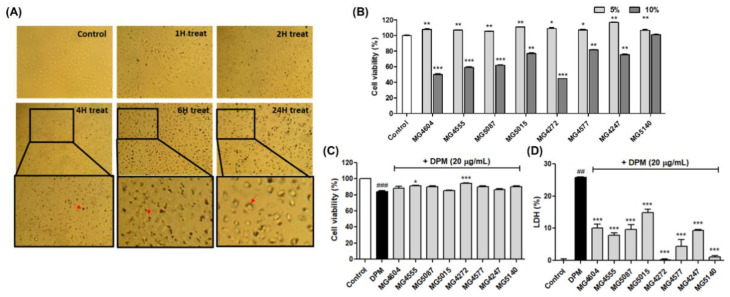
Cell viability and cytotoxicity of DPM treated with or without CFS from *Lactobacillus* and *Streptococcus* strains in NCI-H292 cells. The morphology of NCI-H292 cells, treated at different times (1, 2, 4, 6, and 24 h) of DPM, was observed by the microscope (×20, (**A**)). Red arrows indicate intracellular accumulation of DPM. Cell viability of CFS (**B**) and CFS with DPM (**C**) was measured by MTT assay. Cytotoxicity was assessed by lactate dehydrogenase (LDH) release (**D**). The NCI-H292 cells were pretreated with 5% CFS and then incubated with or only DPM (20 µg/mL) for 24 h. The results indicate the mean ± SEM of three separate experiments. ## *p* < 0.01, and ### *p* < 0.001 compared with control and * *p* < 0.05, ** *p* < 0.01, and *** *p* < 0.001 compared with DPM alone.

**Figure 3 molecules-28-00061-f003:**
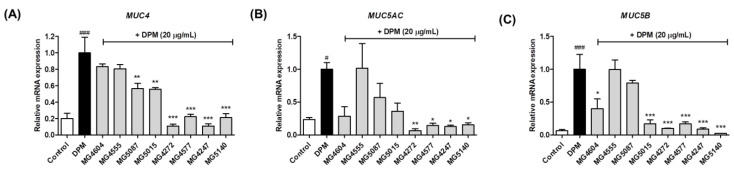
Effect of CFS from Lactobacillus and Streptococcus strains on MUC4 (**A**), MUC5AC (**B**), and MUC5B (**C**) mRNA expression in DPM-induced NCI-H292 cells. The expression of mRNA was determined by qRT-PCR. The NCI-H292 cells were pretreated with 5% CFS and then incubated with or only DPM (20 µg/mL) for 24 h. The mRNA expression was normalized to GAPDH as the internal control. The results indicate the mean ± SEM of four separate experiments. # *p* < 0.05, and ### *p* < 0.001 compared with control and * *p* < 0.05, ** *p* < 0.01, and *** *p* < 0.001 compared with DPM alone.

**Figure 4 molecules-28-00061-f004:**
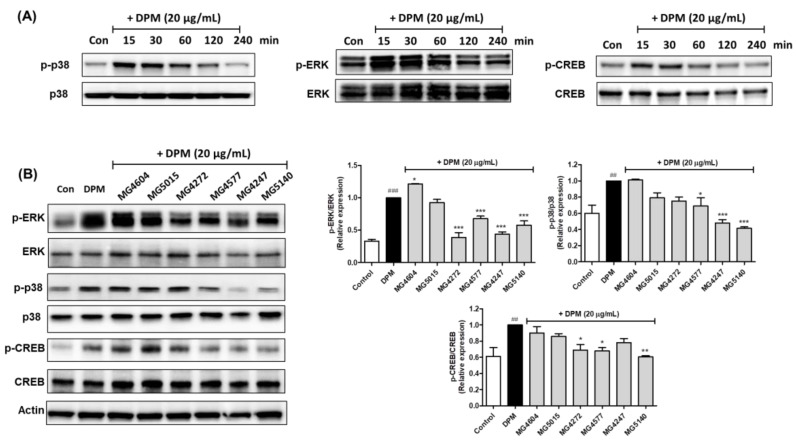
Modulation of the cAMP response element-binding protein (CREB) signaling pathway by CFS from *Lactobacillus* and *Streptococcus* strains in DPM-induced NCI-H292 cells. The NCI-H292 cells were incubated with DPM (20 µg/mL) for various times (15, 30, 60, 120, and 240 min). (**A**). The NCI-H292 cells were pretreated with 5% CFS for 1 h and then incubated with or only DPM (20 µg/mL) for 30 min (**B**). Representative western blot images of protein expression were shown. The protein expression was normalized to β-actin as the internal control. The results indicate the mean ± SEM of three separate experiments. ## *p* < 0.01 and ### *p* < 0.001 compared with control, and * *p* < 0.05, ** *p* < 0.01, and *** *p* < 0.001 compared with DPM alone.

**Figure 5 molecules-28-00061-f005:**
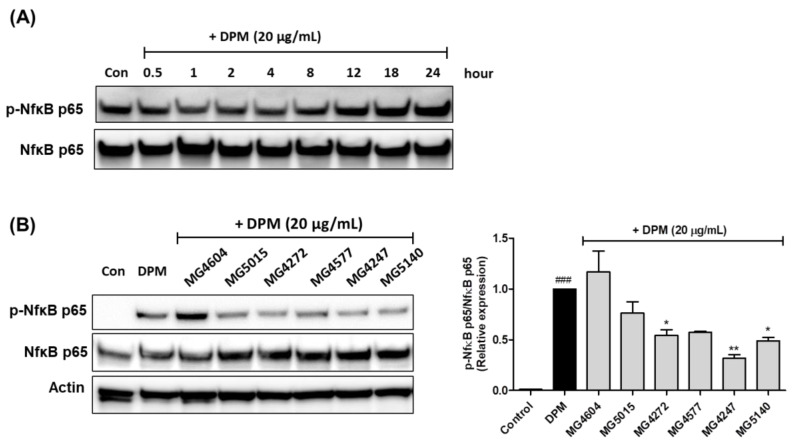
Effect of CFS from *Lactobacillus* and *Streptococcus* strains on phosphorylation of nuclear factor kappa B (Nf-κB) protein expression in DPM-induced NCI-H292 cells. The NCI-H292 cells were incubated with DPM (20 µg/mL) at various times (0.5, 1, 2, 4, 8, 12, 18, and 24 h). (**A**). The NCI-H292 cells were pretreated with 5% CFS 24 h, and then incubated with or only DPM (20 µg/mL) for 30 min (**B**). Representative western blot images of protein expression were shown. The protein expression was normalized to β-actin as the internal control. The results indicate the mean ± SEM of three separate experiments. ### *p* < 0.001 compared with control, and * *p* < 0.05 ** *p* < 0.01, compared with DPM alone.

**Figure 6 molecules-28-00061-f006:**
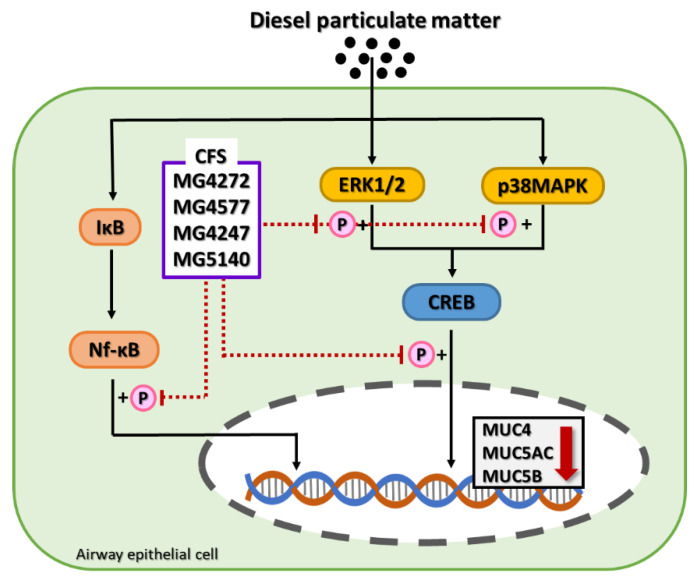
CFS from *L. paracasei* MG4272, MG4577, *L. gasseri* MG4247, and *S. thermophilus* MG5140 inhibits via Nf-κB/CREB pathway, resulting in the down-regulation of mucus production in DPM-induced NCI-H292 cells.

**Table 1 molecules-28-00061-t001:** Primer sequences for qRT-PCR amplification.

Gene ^†^	Sequence (5′-3′)		Product (bp)
*MUC4*	Forward	GACTTGGAGCTCTTTGAGAATGG	139
	Reverse	TGCAATGGCAGACCACAGTCC	
*MUC5AC*	Forward	CTCCTACCAATGCTCTGTA	131
	Reverse	GTTGCAGAAGCAGGTTTG	
*MUC5B*	Forward	GACAGAGACGACAATGAG	154
	Reverse	CCTGATGTTTTCAAAAGTTTC	
*GADPH*	Forward	ACCCACTCCTCCACCTTTG	178
	Reverse	CTCTTGTGCTCTTGCTGGG	

^†^*MUC4*, mucin 4; *MUC5AC*, mucin 5AC; *MUC5B*, mucin 5B; *GAPDH*, glyceraldehyde 3-phosphate dehydrogenase.

## Data Availability

The authors declare that all data and materials support published claims and comply with field standards.

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
