# Peer review of "Cell-Free Supernatant from Lactobacillus and Streptococcus Strains Modulate Mucus Production via Nf-κB/CREB Pathway in Diesel Particle Matter-Stimulated NCI-H292 Airway Epithelial Cells"

_molecules, 2022, doi:10.3390/molecules28010061_

Round 1
Reviewer 1 Report
Comments to Manuscript Molecules-1985099:
1) Page 2, Line 48: A more recent reference regarding probiotics than reference 12, which is more than 10 years old, should be given.
2) Page 2, Line 51: This formulation should be improved.
3) Page 2, Line 57: It should be explained for the induction of airway inflammation what are some equally severe factors as diesel exhaust particles among the various agents mentioned.
4) Page 2, Line 74: The text should refer to (B), not ((C), of figure 1.
5) Page 2, Line 84: This sentence regarding inhibition of NO production does not correspond to the cell viability of CFS with DPM shown in figure 2C.
6) Page 5, Line 155: These metabolites from probiotics should be discussed in more detail.
7) Page 7, Line 237: The availability of these strains for researchers should be mentioned.
Author Response
Authors’ response (in blue) to the Reviewer#1’s comments (in italic black):
Comments to Manuscript Molecules-1985099:
Response:
Thank the anonymous reviewer for thoroughly reading our manuscript and providing helpful comments and suggestions. The detailed responses to major points are listed below:
1) Page 2, Line 48: A more recent reference regarding probiotics than reference 12, which is more than 10 years old, should be given.
Response:
Thank you for your comment, we have changed the reference, “Sanders, M.; Merenstein, D.; Merrifield, C.; Hutkins, R. Probiotics for human use. Nutrition bulletin 2018, 43, 212-225.”, you suggested, P2 Line 48.
2) Page 2, Line 51: This formulation should be improved.
Response:
Thank you for your comment, we have edited the passages.
P2, L50 – “Recently, probiotics has various beneficial effects, such as the prevention of respiratory viruses including COVID-19, anti-allergy, improvement of metabolic disease, and anti-inflammation on skin [14-16].”
3) Page 2, Line 57: It should be explained for the induction of airway inflammation what are some equally severe factors as diesel exhaust particles among the various agents mentioned.
Response:
Thank you for your comment, we explained why we used NCI-H292 cells in this study and have mentioned other agents to induce inflammation.
P2, L56 – “In vitro evaluation has advantages such as more economical, direct access and investigation to cellular components, easier and quicker to perform, and reduction in the number of animals used in research [19]. Also, as a cell model for evaluation of COPD, commonly used NCI-H292 airway epithelial cells from human lungs, which are similar to the response of airway epithelium to cigarette smoke in vivo [19].”
P2, L62 – “To induce airway inflammation among various agents including cigarette smoke extract, fly ash, and carbon black, diesel exhaust particles (DPM) have been reported to induce mucin and inflammation in NCI-H292 airway epithelial cells [20]. ”
4) Page 2, Line 74: The text should refer to (B), not ((C), of figure 1.
Response:
Thank you for your comment, we have edited Figure 1 (please, check the word file).
5) Page 2, Line 84: This sentence regarding inhibition of NO production does not correspond to the cell viability of CFS with DPM shown in figure 2C.
Response:
Thank you for your comment, it is true that we wrote it wrong. So, we have revised it as you suggested in section 2.2.
P2, L90 – “First, cell morphology was observed to set the treatment time of DPM on NCI-H292 cells (Figure 2A). As a result, it was confirmed that DPM was completely penetrated into the NCI-H292cells after 24 h of treatment, so subsequent experiments were conducted at 24 h treatment. Based on results of RAW 264.7 cells, the cytotoxicity of 5 and 10% CFS in NCI-H292 cells, was assessed by MTT assay. In Figure 2B and 2C, 5% CFS of Lactobacillus and Streptococcus strains showed no cytotoxicity (≥86.74%) in the presence or absence of DPM for 24 h. Also, LDH release was measured at 5% CFS. As a result, the 5% CFS of all probiotics significantly inhibited LDH release compared to the DPM-induced controls.”
6) Page 5, Line 155: These metabolites from probiotics should be discussed in more detail.
Response:
Thank you for your comment, we have added the mechanisms of metabolites from probiotics.
P6, L169 – “Metabolites, short chain fatty acids (SCFA) and microbe associated molecular patterns (MAMPs) derived from the probiotics, suppresses loss of immune homeostasis and inflammatory response caused by excessive DPM exposure by promoting tight-junction proteins to repair the intestinal epithelial barrier function or prevents COPD by directly influencing pulmonary immune homeostasis [21].”
7) Page 7, Line 237: The availability of these strains for researchers should be mentioned.
Response:
Thank you for your comment, the availability of data and materials was mentioned in Data Availability Statement, P10, L345.

Reviewer 2 Report
In this study, the authors investigated the effects of Cell-Free Supernatants (CFS) prepared from Lactobacillus and Streptococcus on nitro oxide (NO) production in RAW264.7, and proteins associated with mucus production in NCI-H292 cells. It is very interesting. However, the real effect may be the metabolites secreted by the probiotics rather than the probiotics themselves The authors should describe it clearly. Here are my comments:
1. The authors should rationalize the possibility of clinical application, whether they intended to use probiotics themselves to protect the airway, or to prepare metabolites secreted from probiotics as dietary supplements. If the former, how can probiotics grow safely in the airways without any inflammation? If it is the latter, the relevant descriptions should be written clearly in the full text.
2. Each experiment should provide the case number, and it is from several independent experiments or a single experiment.
3. MRS components are complex, how to rule out the beneficial effect of MRS components in CFS? Should you include MRS broth as a control?
4. Abstract, Discussion, Conclusions, and many other places confuse the metabolites with the role of probiotics. Many places in the full text, including the title, need to be revised.
5. Figure 1, legend (A), (B), (C) is incorrect.
6. The species name should be indicated when NCI-H292 cells appear for the first time.
7. In line 315, "may be used as safe dietary supplements to improve COPD.", such as comment 1, please clarify whether this conclusion is correct.
8. In the diagram of the conclusion, it is not the probiotics themselves but the secreted metabolites that act, which should be revised.
Author Response
Authors’ response (in blue) to the Reviewer#2’s comments (in italic black):
In this study, the authors investigated the effects of Cell-Free Supernatants (CFS) prepared from Lactobacillus and Streptococcus on nitro oxide (NO) production in RAW264.7, and proteins associated with mucus production in NCI-H292 cells. It is very interesting. However, the real effect may be the metabolites secreted by the probiotics rather than the probiotics themselves. The authors should describe it clearly. Here are my comments:
Response:
thank the anonymous reviewer for thoroughly reading our manuscript and providing helpful comments and suggestions. The detailed responses to major points are listed below:
- The authors should rationalize the possibility of clinical application, whether they intended to use probiotics themselves to protect the airway, or to prepare metabolites secreted from probiotics as dietary supplements. If the former, how can probiotics grow safely in the airways without any inflammation? If it is the latter, the relevant descriptions should be written clearly in the full text.
Response:
Thank you for your comment. If probiotics enter the lungs directly and grow, as you mentioned, inflammation will occur and have a critical effect on the human body. The gut-lung axis has been described in many reports, and its mechanism of action is as follows:
P6, L169 – “Metabolites, short chain fatty acids (SCFA) and microbe-associated molecular patterns (MAMPs) derived from the probiotics, suppresses loss of immune homeostasis and inflammatory response caused by excessive DPM exposure by promoting tight-junction proteins to repair the intestinal epithelial barrier function or prevents COPD by directly influencing pulmonary immune homeostasis [21].”
Also, the reason we tested using CFS from Lactobacillus and Streptococcus in a cell model was to confirm the effect of metabolites produced in airway epithelial cells when probiotics were consumed later in animal or clinical trial studies.
- Each experiment should provide the case number, and it is from several independent experiments or a single experiment.
Response:
Thank you for your comment, we have provided the figure legends of each experiment to “The results indicate the mean ± SEM of three separate experiments” or“The results indicate the mean ± SEM of four separate experiments”.
- MRS components are complex, how to rule out the beneficial effect of MRS components in CFS? Should you include MRS broth as a control?
Response:
Thank you for your comment. As you said, since MRS is complex, we agree that the efficacy of CFS should be conducted reliably through comparison with the control of MRS treatment. Our result is the addition of the same amount of MRS (5%) to the control group. We added these contents to the material and methods.
- Abstract, Discussion, Conclusions, and many other places confuse the metabolites with the role of probiotics. Many places in the full text, including the title, need to be revised.
Response:
Thank you for your comment, we have revised it in full text and changed the title “Cell-Free Supernatant from Lactobacillus and Streptococcus strains modulate mucus production via Nf-κB/CREB pathway in diesel particle matter-stimulated NCI-H292 airway epithelial cells”.
P2, L65 – “In this study, we confirmed that inhibition of cell-free supernatant (CFS) from Lactobacillus and Streptococcus strains on inflammation via NO production in LPS-induced RAW264.7 cells. Moreover, effect of CFS from Lactobacillus and Streptococcus strains on the expression of mucin-related genes and proteins in NCI-H292 cells treated with DPM was investigated.”
- Figure 1, legend (A), (B), (C) is incorrect.
Response:
Thank you for your comment, we have revised Figure 1 (Please, check the word file).
- The species name should be indicated when NCI-H292 cells appear for the first time.
Response:
Thank you for your comment, we have added the origin of NCI-H292 in P2, L60.
- In line 315, "may be used as safe dietary supplements to improve COPD.", such as comment 1, please clarify whether this conclusion is correct.
Response:
Thank you for your comment, we have revised the passage.
P9, L332 – “~ may be used as safe dietary supplements to improve airway inflammation, which needs to be proven in further animal studies.”
- In the diagram of the conclusion, it is not the probiotics themselves but the secreted metabolites that act, which should be revised
Response:
Thank you for your comment, we have revised the diagram, Figure 6.

Reviewer 3 Report
Lactobacillus and Streptococcus strains suppress mucus pro- 2 duction via Nf-κB/CREB pathway in diesel particle matter- 3 stimulated human airway epithelial cells
Major points:
As general comment, this study complements a previous one from the authors that measures the effects on RAW 264.7 and RBL-2H3 mast cells of CFS from some of the same strains of the current study and others new. This is not explained in the introduction. Thus, some of the experiments of the current manuscript done for RAW 264.7 are not complete because the authors have deduced some of the effects from the previous work. In the present form, the manuscript cannot be accepted. The authors must complete the experiments that have to be done and change the focus of the study, rewriting for this, the introduction and the discussion.
The current manuscript has relevance due to the described effects on NCI- 310 H292 airway epithelial cells and that is what the authors have to bold.
Besides, I recommend changing the title and detail the strains that induce the suppression.
Other major points:
. To confirm the inhibition of inflammation by the bacterial supernatants in RAW 264.7 cells, cytokines such as TNF-alpha and IL-6 should be measured. The authors refer to the previous manuscript where they assayed inflammatory cytokines with some of the bacterial strains, but not with all the ones used for the present manuscript.
NO measurement is interesting as a mediator of inflammation, but it is not enough.
Why LDH determination has not been done for RAW 264.7 exposed to the bacterial CFS? It should be done.
Minor points:
Please rephrase: Line 50-51:
Recently, probiotics has various beneficial effects, such as in respiratory viruses infections including COVID-19, prevention of allergies, metabolic disease, and skin 51 inflammation [14-16].
Line 57-62: please correct: For induce airway inflammation among various agents, diesel exhaust particles (DPM) have been reported to induce mucin and inflammation in NCI-H292 airway epithelial cells [19]. Therefore, to confirm the mechanism of inhibition of airway inflammation by Lactobacillus and Streptococcus strains, we confirmed NO production in LPS-induced RAW264.7 cells and treated NCI-H292 cells with DPM to confirm the expression of mucin-related genes and proteins. Rephrase. It is not well expressed what you have done and what for.
Please revise that all the names of microorganisms are in italics.
Figure1: ¿Compared to DPM alone? In this figure, DPM is not present.
Figure 1. Effect of CFS from Lactobacillus and Streptococcus strains on cell viability without (A) or 73 with LPS (A), and production of NO with LPS (C) in RAW 264.7 cells. The RAW 264.7 cells were 74 pretreated with 2.5 and 5% of CFS and then incubated with or without only LPS (0.5 µg/mL) for 24 75 Molecules 2022, 27, x FOR PEER REVIEW 3 of 11 h. The results indicate the mean ± SEM of three separate experiments. # p < 0.05, and ### p < 0.001 76 compared with control and *p < 0.05, **p < 0.01, and ***p < 0.001 compared with DPM alone.
Figure 2: please improve the quality of the photographs.
Lines: 114-121: please rephrase for clarity. Were the CFS of Lactobacillus and Streptococcus pretreated, or were the cells?
Author Response
Authors’ response (in blue) to the Reviewer#3’s comments (in italic black):
Major points:
As general comment, this study complements a previous one from the authors that measures the effects on RAW 264.7 and RBL-2H3 mast cells of CFS from some of the same strains of the current study and others new. This is not explained in the introduction. Thus, some of the experiments of the current manuscript done for RAW 264.7 are not complete because the authors have deduced some of the effects from the previous work. In the present form, the manuscript cannot be accepted. The authors must complete the experiments that have to be done and change the focus of the study, rewriting for this, the introduction and the discussion.
Response:
thank the anonymous reviewer for thoroughly reading our manuscript and providing helpful comments and suggestions. The detailed responses to minor points are listed below:
The current manuscript has relevance due to the described effects on NCI- H292 airway epithelial cells and that is what the authors have to bold.
Besides, I recommend changing the title and detail the strains that induce the suppression.
Response:
Thank you for your comment, we explained why we used NCI-H292 cells in this study. And, we changed the title to “Cell-Free Supernatant from Lactobacillus and Streptococcus strains modulate mucus production via Nf-κB/CREB pathway in diesel particle matter-stimulated NCI-H292 airway epithelial cells”.
P2, L56 – “In vitro evaluation has advantages such as more economical, direct access and investigation to cellular components, easier and quicker to perform, and reduction in the number of animals used in research [19]. Also, as a cell model for evaluation of COPD, commonly used NCI-H292 airway epithelial cells from human lungs, which are similar to the response of airway epithelium to cigarette smoke in vivo [19].”
Other major points:
To confirm the inhibition of inflammation by the bacterial supernatants in RAW 264.7 cells, cytokines such as TNF-alpha and IL-6 should be measured. The authors refer to the previous manuscript where they assayed inflammatory cytokines with some of the bacterial strains, but not with all the ones used for the present manuscript.
Response:
Thank you for your comment. In ref. [26], MG4247, MG4577, and MG4272 were measured, and IL6 and TNF-α for the remaining strains were shown in Figure S1. Figure S1 is as follows: (Please, check the word file).
NO measurement is interesting as a mediator of inflammation, but it is not enough.
Response:
Thank you for your comment. As mentioned above, we also measured IL-6 and TNF-α, agreeing with you that NO alone is not enough in P6, L192.
Why LDH determination has not been done for RAW 264.7 exposed to the bacterial CFS? It should be done.
Response:
Thank you for your comment. Based on your opinion, we have assessed LDH releases of bacterial CFS on RAW 264.7 cells shown in Figure 1C (Please, check the word file).
P2, L75– “Moreover, LDH release of 5% CFS on LPS-induced RAW 264.7 cells was measured shown in Figure 1B. All CFS decreased LDH release compared to LPS-induced control.”
Minor points:
Please rephrase: Line 50-51:
Recently, probiotics has various beneficial effects, such as in respiratory viruses infections including COVID-19, prevention of allergies, metabolic disease, and skin 51 inflammation [14-16].
Response:
Thank you for your comment, we have edited the passages.
P2, L50 – “Recently, probiotics has various beneficial effects, such as the prevention of respiratory viruses including COVID-19, anti-allergy, improvement of metabolic disease, and anti-inflammation on skin [14-16].”
Line 57-62: please correct:
For induce airway inflammation among various agents, diesel exhaust particles (DPM) have been reported to induce mucin and inflammation in NCI-H292 airway epithelial cells [19]. Therefore, to confirm the mechanism of inhibition of airway inflammation by Lactobacillus and Streptococcus strains, we confirmed NO production in LPS-induced RAW264.7 cells and treated NCI-H292 cells with DPM to confirm the expression of mucin-related genes and proteins. Rephrase. It is not well expressed what you have done and what for.
Response:
Thank you for your comment. We tried to describe the reason for using DPM to NCI-H292, and the purpose of this study. According to your opinion, to clarify the meaning, we have rephrased the passages.
P2, L65 – “In this study, we confirmed that inhibition of cell-free supernatant (CFS) from Lactobacillus and Streptococcus strains on inflammation via NO production in LPS-induced RAW264.7 cells. Moreover, effect of CFS from Lactobacillus and Streptococcus strains on the expression of mucin-related genes and proteins in NCI-H292 cells treated with DPM was investigated.”
Please revise that all the names of microorganisms are in italics.
Response:
Thank you for your comment, we have revised in P9, L327, and throughout the section of Materials and Methods.
Figure1: ¿Compared to DPM alone? In this figure, DPM is not present.
Figure 1. Effect of CFS from Lactobacillus and Streptococcus strains on cell viability without (A) or with LPS (A), and production of NO with LPS (C) in RAW 264.7 cells. The RAW 264.7 cells were pretreated with 2.5 and 5% of CFS and then incubated with or without only LPS (0.5 µg/mL) for 24 Molecules 2022, 27, x FOR PEER REVIEW 3 of 11 h. The results indicate the mean ± SEM of three separate experiments. # p < 0.05, and ### p < 0.001 76 compared with control and *p < 0.05, **p < 0.01, and ***p < 0.001 compared with DPM alone.
Response:
Thank you for your comment, we have rewritten the legend of Figure 1.
“Figure 1. Effect of CFS from Lactobacillus and Streptococcus strains on cell viability without (A) or with LPS (A), lactate dehydrogenase (LDH) release (B), and production of nitro oxide (NO) with LPS (C) in RAW 264.7 cells. The RAW 264.7 cells were pretreated with 2.5 and/or 5% of CFS for 1 h, and then incubated with or without only LPS (0.5 µg/mL) for 24 h. The results indicate the mean ± SEM of three separate experiments. ### p < 0.001 compared with control and *p < 0.05, **p < 0.01, and ***p < 0.001 compared with LPS alone.”
Figure 2: please improve the quality of the photographs.
Response:
Thank you for your comment, we agree with your comments. We increased the size of the photographs in figure 2A and improved the resolution by increasing it to 600 dpi: (Please, check the word file).

Round 2
Reviewer 2 Report
The conclusion is still not accurate enough in the revised MS. I have no more comments.
Author Response
Authors’ response (in blue) to the Reviewer#2’s comments (in italic black):
The conclusion is still not accurate enough in the revised MS. I have no more comments:
Response:
Thank the anonymous reviewer for thoroughly reading our manuscript and providing helpful comments and suggestions.
Reviewer 3 Report
The authors have improved the manuscript, however there is still room for improvement.
Major requirements:
. I still miss in the introduction mentions of the previous work and other works supporting the model used by the authors.
. IL-6 and TNF-alpha measurements should be shown as figure in results, and the differences among CFS effects on RAW 264.7 and NCI-H292 discussed.
Minor points:
. Please revise the following paragraph. Try not to repat shown and the new phrase needs correction: lines 73 and forward.
The cytotoxicity of 2.5 and 5% CFS in RAW 264.7, was established by MTT assay (Figure 1A and 1B). The results showed that 5% CFS of Lactobacillus and Streptococcus 74 strains showed no cytotoxicity (≥100%) in the presence or absence of LPS for 24 h. Moreover, LDH release of 5% CFS on LPS-induced RAW 264.7 cells was measured and shown in Figure 1B. All CFS decreased LDH release compared to LPS-induced control. Therefore, NO production was evaluated at 5% CFS, a concentration without cytotoxicity. As shown in Figure 1C, the CFS of all probiotics remarkably inhibited NO production compared to the LPS-induced controls.
. Please rephrase paragraph between lines 207 and 220. The good and coherent results are not well described and justified.
. line 227: I would not say that these probiotics, it is said, alive microorganisms, are functional foods.
Author Response
Authors’ response (in blue) to the Reviewer#3’s comments (in italic black):
The authors have improved the manuscript, however there is still room for improvement.
Response:
thank the anonymous reviewer for thoroughly reading our manuscript and providing helpful comments and suggestions. The detailed responses to points are listed below:
Major requirements:
I still miss in the introduction mentions of the previous work and other works supporting the model used by the authors.
Response:
If you are talking about the introductory mention of our previous work, ref. [29], we have added conclusions about that paper in L221. And, an overview of the cell lines and treatment materials used to rationalize our in vitro model is given between L56 and L63. In addition, recent literature using this model was further written in L64.
P7, L221 – “In our previous reports, L. paracasei MG4272, MG4577, L. gasseri MG4247 attenuate allergic inflammatory response in mast cell by modulating signal transducer and activator of transcription 6 (STAT6) phosphorylation [29].”
P2, L64 – “Based on this considerable evidence, it was recently reported that the airway inflammation of saponin was confirmed using an in vitro model in which DPM was treated with NCI-H292 [21].”
IL-6 and TNF-alpha measurements should be shown as figure in results, and the differences among CFS effects on RAW 264.7 and NCI-H292 discussed.
Response:
Unfortunately, we are only providing results for some strains, so we do not think we can present them in the text. In addition, rather than attempting to explain the differences between the inflammatory cytokines identified in RAW 264.7 cells and NCI-H292, we discussed that the inflammatory cytokines identified in RAW 264.7 may indirectly affect airway inflammation, in line 193-197.
Minor points:
Please revise the following paragraph. Try not to repat shown and the new phrase needs correction: lines 73 and forward.
The cytotoxicity of 2.5 and 5% CFS in RAW 264.7, was established by MTT assay (Figure 1A and 1B). The results showed that 5% CFS of Lactobacillus and Streptococcus 74 strains showed no cytotoxicity (≥100%) in the presence or absence of LPS for 24 h. Moreover, LDH release of 5% CFS on LPS-induced RAW 264.7 cells was measured and shown in Figure 1B. All CFS decreased LDH release compared to LPS-induced control. Therefore, NO production was evaluated at 5% CFS, a concentration without cytotoxicity. As shown in Figure 1C, the CFS of all probiotics remarkably inhibited NO production compared to the LPS-induced controls.
Response:
We have revised the paragraph in P2, L75.
Please rephrase paragraph between lines 207 and 220. The good and coherent results are not well described and justified.
Response:
As per your comment, we have rephrased between lines 209 and 221.
line 227: I would not say that these probiotics, it is said, alive microorganisms, are functional foods.
Response:
We have deleted “probiotics” in line 230.